# NORESQA : A Framework for Speech Quality Assessment using Non-Matching References

**Pranay Manocha**[*]
Department of Computer Science
Princeton University
Princeton, NJ
pmanocha@cs.princeton.edu

**Buye Xu**
Facebook Reality Labs Research
Redmond, WA
xub@fb.com

**Anurag Kumar**
Facebook Reality Labs Research
Redmond, WA
anuragkr@fb.com

## Abstract

The perceptual task of speech quality assessment (SQA) is a challenging task for machines to do. Objective SQA methods that rely on the availability of the corresponding clean reference have been the primary go-to approaches for SQA. Clearly, these methods fail in real-world scenarios where the ground truth clean references are not available. In recent years, non-intrusive methods that train neural networks to predict ratings or scores have attracted much attention, but they suffer from several shortcomings such as lack of robustness, reliance on labeled data for training and so on. In this work, we propose a new direction for speech quality assessment. Inspired by human's innate ability to compare and assess the quality of speech signals even when they have non-matching contents, we propose a novel framework that predicts a subjective *relative* quality score for the given speech signal with respect to *any provided reference* without using any subjective data. We show that neural networks trained using our framework produce scores that correlate well with subjective mean opinion scores (MOS) and are also competitive to methods such as DNSMOS [1], which explicitly relies on MOS from humans for training networks. Moreover, our method also provides a natural way to embed quality-related information in neural networks, which we show is helpful for downstream tasks such as speech enhancement.

## 1 Introduction

Speech quality assessment is critical for designing and developing a wide range of real-world audio and speech applications, such as, Telephony, VoIP, Hearing Aids, Automatic Speech Recognition, Speech Enhancement etc. Clearly, the gold standard for SQA is the evaluation of speech recordings by humans. However, these subjective evaluations are not scalable and can be immensely time-consuming and costly, as they often need to be repeated tens or hundreds of times for every recording. Several objective methods for SQA have been developed to address this problem, PESQ [2], POLQA [3], VISQOL [4], HASQI [5], DPAM [6] and CDPAM [7]. These methods are *intrusive* or *full-reference* by definition as they are designed to produce a quality score or rating by comparing the corrupted speech signal to its clean reference. However, they suffer from three critical drawbacks. *First*, the requirement of a paired clean reference for quality assessment limits their

---

[*]Work done during internship at Facebook Reality Labs Research

35th Conference on Neural Information Processing Systems (NeurIPS 2021).

applicability to real-world scenarios as the paired clean reference is likely not available in those cases. *Second*, these methods have acknowledged shortcomings such as sensitivity to perceptually invariant transformations [8], therefore hindering stability in more diverse tasks such as speech enhancement. *Lastly*, these metrics are non-differentiable and cannot be directly leveraged as training objectives in the context of neural networks.

To address these problems, a recent trend has been to develop *non-intrusive* [9] methods using neural networks [1, 10–20]. In most cases, the primary approach is to train a model to predict objective (e.g., PESQ) and/or subjective (e.g., MOS) scores. However, generalization to unseen perturbations and tasks remains a concern [21], and most methods have not found wide-spread uses for SQA. Given that matching human subjective ratings is the ultimate motivation, some recent works try to train neural networks directly on MOS scores [20, 14, 1]. DNSMOS [1], in particular, trains neural networks on a very large-scale MOS database. However, collecting such a dataset is an uphill task and requires considerable resources. For e.g., one requires uniformity with respect to hardware (headphones/speakers), listening environments etc., among hundreds and thousands of raters to ensure that ratings are consistent. Otherwise, a significant amount of noise can creep into the dataset, making it unreliable for training. In DNSMOS [1], almost half of the recordings have a standard deviation of more than 1 (MOS $\pm 1$ ), which poses challenges in training robust models due to noisy labels.

Lastly, we would like to point out an inherent challenge of the conventional formulation of any non-intrusive metric. The problem might lie in the *lack of reference* itself. While training any model on subjective scores, we expect the model to implicitly learn the distribution of references that are consciously or unconsciously used by human listeners. These references can be highly varied and strongly influenced by each individual's past experience and even the mood when participating in the evaluations. Learning such a distribution can be an extremely hard problem, especially when no specific constraints on the distribution of clean reference are provided to the model during training.

In this work, we propose a novel and alternate framework for speech quality assessment. Instead of a completely reference-free approach, our framework relies on random non-matching references (NMRs) of known qualities, and is designed to provide a relative assessment of speech quality with respect to the NMRs. The inspiration for the approach comes from human's ability to do the same. Given two completely random speech recordings, it is highly likely that a human would be able to compare them with respect to quality irrespective of the actual speech content. This innate ability to compare the quality of two speech recordings in an "unsupervised" setting (or non-matching conditions) holds true even when the recordings contain different speakers, languages, words, and so on. Moreover, comparative tests are relatively easier for humans than absolute rating, and relative scores tend to have lower variance and less noise.

Motivated by the above points, we propose NORESQA - *NOn-matching REference based Speech Quality Assessment*. Within this framework, we propose to learn neural networks that can predict a *relative quality score for a given speech recording with respect to any provided reference*. A few key characteristics of NORESQA are: (i) unlike full-reference objective metrics, it remains usable in real world situations by relying on NMRs which are readily available; (ii) it addresses the problem of lack of reference in non-intrusive methods by providing an NMR, thereby providing the necessary grounding for the model to learn and predict acoustic quality differences; (iii) pairwise comparisons are expected to have lower variance and less noise than absolute ratings (e.g MOS); and (iv) any learning within this framework is "unsupervised" in the sense that we do not require any manually labeled dataset for training models.

To summarize the *key contributions* of the paper: (1) we propose a novel framework for speech quality assessment that relies on NMRs; (2) we propose methods to train neural networks within this framework using multi-task multi-objective learning; and (3) we evaluate our framework comprehensively through several subjective and objective evaluations as well as exploring its utility in a downstream task like speech enhancement.

## 2   Related Work

**Non-Intrusive Methods.** Some of the earliest non-intrusive methods were based on complex hand-crafted, rule-based systems [22–25]. Although they are automatic and interpretable, they tend to be task-specific, and do not generalize well. Moreover, these methods are non-differentiable which limits their uses within deep learning frameworks. To overcome the last concern, various neural network based methods have been developed [1, 10–18]. However, the issue of task-specificity and

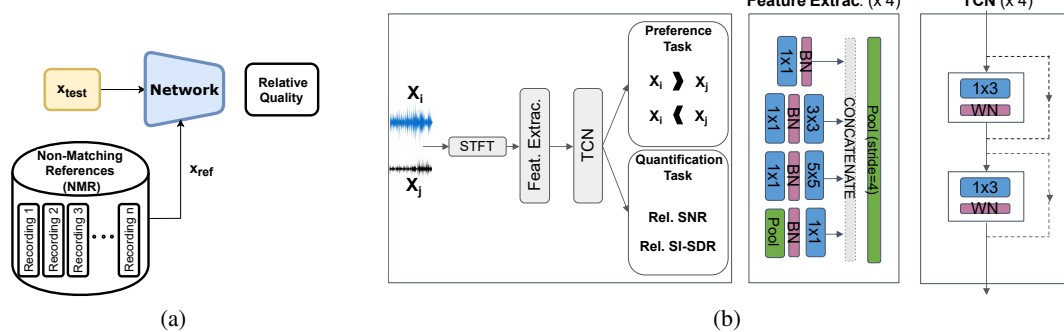

(a)                       (b)

Figure 1: *Left:*(a) **Proposed NORESQA framework** takes a test recording, and a randomly chosen NMR from a set of NMRs and predicts the relative quality. *Right*: **NORESQA Architecture** (b) Our network takes two non-matched recordings ($x_\mathtt{i}$ and $x_\mathtt{j}$) and outputs: (i) which recording is cleaner (*preference-task*); and (ii) relative quality difference using SI-SDR and SNR (*quantitative-task*).

generalization remained. To overcome this, researchers proposed to train models directly on a dataset of human judgment scores [1, 14, 20]. Reddy et al. [1] used a multi-stage self-teaching model [26] to learn quality in the presence of noisy ratings. Nonetheless, non-intrusive metrics lag behind intrusive metrics in terms of correlation to human listening evaluations and adoption in practical cases.

**Vision.** Researchers in the vision community have also explored the field of no-reference image quality assessment. Various works have looked at approaches using learning to rank [27–29] rather than training on absolute ratings. However, to the best of our knowledge, no prior work has explored a framework that relies on random NMRs to provide a relative assessment of quality.

**Multi-task learning for audio quality.** Multi-task learning (MTL) [30] has been beneficial to many speech applications [31–34]. MTL aims to leverage useful information contained in multiple related tasks to help improve the generalization performance on all tasks. Serra et al. [35] proposed SESQA learned from various objectives like MOS, pairwise ranking, score consistency, and other metrics like PESQ and STOI etc,. However, their approach uses paired clean and noisy recordings for training and also requires access to a dataset of subjective ratings. In contrast, our proposed quality assessment framework works in a "unsupervised" setting and does not require any labeled data. We leverage MTL to learn a preference and a quantification task for quality assessment.

## 3 NORESQA Framework

Our framework, NORESQA , is designed to assess the quality of a given speech recording using Non-Matching References (NMRs). This novel framework is a *generic idea*, and within it, one can develop a variety of methods to produce a relative quality score (referred to as NORESQA score) for a test input, $x_{test}$, with respect to any given reference, $x_{ref}$. We propose a deep neural network for NORESQA and represent it by the function $\text{NORESQA} = \mathcal{N}(x_{test}, x_{ref})$. Given that we do not rely on any human-labeled data, the crucial components of the framework include designing tasks and objective functions that can help learn a quality score. This is along the lines of self-supervised learning [36–38], where supervised tasks are designed on unlabeled data for training neural networks.

### 3.1 Overview

The NORESQA framework takes in two recordings as inputs - test recording $x_\mathtt{test}$ and another randomly chosen recording $x_\mathtt{ref}$. Fig 1(a) is a simple illustration of the framework. In our current approach, NORESQA has the following properties: (i) Non-negative (by design) $\mathcal{N}(x_{test}, x_{ref}) \geq 0$; (ii) Monotonic (by design) if $\mathcal{N}(x_{test1}, x_{ref}) \geq \mathcal{N}(x_{test2}, x_{ref})$, then $\mathtt{m}(x_{test1}) \leq \mathtt{m}(x_{test2})$, where $\mathtt{m}$ is any quality assessment measure as defined in Section 3.3; (iii) a way to predict "sign", which helps differentiate $\mathcal{N}(x_a, x_b)$ vs. $\mathcal{N}(x_b, x_a)$. We do not enforce other metric properties [39, 40] to allow flexibility in defining tasks and objectives for training the neural networks. Moreover, even human judgment of similarity may not constitute a metric [41], and hence there is no pertinent reason which necessitates NORESQA to have metric properties.

The network architecture is shown in Fig 1(b). The neural network model is designed to detect and quantify degradation at the frame-level (frame $\approx 32$ ms of audio). To learn quality, we propose a multi-task multi-objective training mechanism in which the network aims to classify which input is better, as well as "quantify by how much", in terms of two measures, scale-invariant signal to distortion ratio (SI-SDR) and signal-to-noise ratio (SNR). Additionally, our framework outputs frame-level judgments, by which we can detect and quantify which frames are degraded in quality with respect to the NMRs.

## 3.2 Architecture

NORESQA's architecture (shown in Fig 1(b)) comprises of three key components: a *feature-extraction* block, a *temporal-learning* block and *task-specific* heads. The feature-extraction block learns representations while maintaining the temporal structure of the signal. These representations are then fed to the temporal-learning block to learn long-term dependencies using temporal convolutional networks (TCN). The resulting embeddings are then fed to the task-specific heads to produce frame-wise outputs. Note that the architecture is an instance of shared parameters for the two inputs, in other words both inputs are processed by the same network.

**Feature-Extraction Block.** The inputs to the network are 3-second audio excerpts, represented by their Short-Time Fourier Transform (STFT). We stack together the magnitude and phase of the STFT as two channels, leading to a $2 \times T \times F$ dimensional representation of every audio recording where $T$ is the number of frames and $F$ is number of frequency bins. More specifically, STFTs are computed using a hamming window of size 32 ms with 50% overlap. Only the 256 positive frequencies without the zeroth bin are used. We use the Inception [42] architecture in the feature extraction block. Please refer to the supplementary material for details of the feature-extraction block.

**Temporal Learning Block.** The temporal learning block is aimed at learning long-term temporal information from frame-level representations generated by the feature-extraction block. We use Temporal Convolutional Networks (TCNs) [43] in this block. While the phonetic speech content can change per frame, the acoustics (recording conditions, background noise, distortions, etc) necessitates capturing long-term history for quality assessment. A cascade of TCNs offers a flexible way to learn both short-term (local) and long-term information. We request the readers to refer to supplementary material for precise details of the TCNs used in this block.

The learned weights across the first two blocks are shared between the two inputs to our model. The outputs of the temporal learning block for both inputs are concatenated, and then fed into each of the task-specific heads. Next, we describe the multi-task multi-objective learning mechanism:

## 3.3 Multi-task and multi-objective learning

The NORESQA network architecture described above is trained through a multi-task multi-objective approach. Since we do not have any perceptual labels, we rely on two signal processing measures, Signal-to-Noise Ratio (SNR) and Scale-Invariant Signal to Distortion Ratio (SI-SDR) to compare the quality of the two inputs.

SNR is measured as the ratio of the signal power to the noise power and is primarily meant only for additive noises. Consider a mixture signal $x$, $x = s + n \in \mathbb{R}^L$ where $s$ is the clean signal and $n$ is the noise signal, then

$$\text{SNR} = 10 \log_{10} \left( \frac{||s||^2}{||s - x||^2} \right) \tag{1}$$

$10 log_{10}()$ factor measures SNR in dB-scale, and a higher SNR implies better signal quality. Yuan et al. [44] also showed that SNR as a distance metric had better properties than conventional metrics (like Euclidean distance).

SI-SDR is a measure that was introduced to evaluate performance of speech processing algorithms. It is invariant to the scale of the processed signal and can be used to quantify quality in diverse cases, including additive background noises as well as other distortions [45].

Given a noisy mixture $x$ and it's clean counterpart $s$, the SI-SDR is defined as:

$$\text{SI-SDR} = 10 \log_{10} \left( \frac{||\alpha s||^2}{||\alpha s - x||^2} \right), \text{ where } \alpha = \arg\min_{\alpha} ||\alpha s - x|| = \frac{x^T s}{||s||^2} \tag{2}$$

This property of scale invariance is very important since both speech quality and intelligibility to a large extent are invariant to scaling [46]. SI-SDR has been shown to be an effective approach for quality measurement [47, 48]. The motivation behind relying on SNR and SI-SDR is that these are two of the most fundamental and generic measures to quantify the quality of a signal. They have been used extensively in training and evaluating audio source separation and speech enhancement algorithms [49–51, 47, 48], and also correlate well with human perception on various realistic tasks [52, 53].

We use the SNR and SI-SDR measures as "quality labels" for all speech recordings, and the objective of the network is to solve two tasks. Given two audio inputs, the network solves: (a) a *Preference*

*Task*: which input audio is of better quality?; and (b) a *Quantification Task*: what is the quality difference between the two audio inputs w.r.t their "quality labels"? Separating the learning task into two separate tasks is important is because the inputs to our model are non-matching pairs. If we use signed intervals, it might be non-trivial for the model to learn to discard speech content differences and learn quality features. Disentangling the learning through two separate tasks makes it easier for the mode to focus on quality attributes. It also makes the model easier to use - in the sense that if only preference results are needed or if only absolute quality difference is required, the appropriate head can be used while discarding the unused one. Lastly, the framework can be easily extended through the separate preference and quantification tasks. For example, we can have more than 2 inputs and the preference task can be designed to predict a ranked order in terms of quality.

Note that both of these tasks are relative assessments as opposed to absolute quality measurements. Human listeners find relative assessment an easier task than absolute quality ratings [54] and relative subjective scores are typically more robust than absolute scores [55]. Relative scores tend to give more repeatable results [56] and are likely to have lower variance and bias. Hence, we argue that neural networks trained on these relative assessments will likely lead to better generalization. Moreover, just as humans can compare the quality between non-matching recordings, our training mechanism also follows the same strategy.

**Preference Task.** The preference task is formulated as a binary classification problem. Let $\mathbf{x}_{ij} = (x_i, x_j)$ be an *ordered* pair input to the network, with $x_i$ as first input and $x_j$ as second input. Let $sdr_{x_i}$ and $sdr_{x_j}$ be the SI-SDRs of $x_i$ and $x_j$ respectively. The label $\mathbf{y}_{ij}$ for $\mathbf{x}_{ij}$ is a 2 dimensional, one-hot vector, with $\mathbf{y}_{ij} = [1, 0]$ if $sdr_{x_i} > sdr_{x_j}$, otherwise $\mathbf{y}_{ij} = [0, 1]$. The preference-task head is a small, fully convolutional network and outputs a distribution of which input is cleaner at frame-level. The frame-level distributions are temporally averaged to produce the recording level distribution $\mathbf{p}_{ij}$ which are then used in the standard cross-entropy function to compute loss for this task.

$$L_P(\mathbf{x}_{ij}, \mathbf{y}_{ij}) = \sum_{k=1}^{2} -y_{ij}^k \log(p_{ij}^k) \qquad (3)$$

**Quantification Task.** This task is designed to quantify the quality differences between $x_i$ and $x_j$. Let $snr_{x_i}$ and $snr_{x_j}$ be SNRs of $x_i$ and $x_j$ respectively. This task consists of two output heads, one for predicting quality differences in terms of SNR, $\Delta snr_{ij} = |snr_{x_i} - snr_{x_j}|$, and the other for SI-SDR $\Delta sdr_{ij} = |sdr_{x_i} - sdr_{x_j}|$. Architecturally both heads are identical, the details of which are provided in the supplementary material. Trivially, one can formulate $\Delta snr$ or $\Delta sdr$ prediction as regression problems [14, 35]. Here, we formulate them as classification problems. By applying tricks from deep neural network based classification methods, we are able to learn much more robust models.

Let $\Delta sdr_{max} = \max_{ij} |sdr_{x_i} - sdr_{x_j}|$ be the maximum absolute difference in SI-SDR among all pairs of $x_i$ and $x_j$ in the dataset. We divide the range of $\Delta sdr$, from 0 to $\Delta sdr_{max}$, into $K$ equally spaced bins. Each bin is a class and the label for $\mathbf{z}_{ij}$ for input $\mathbf{x}_{ij}$ is $k^{th}$ class if $\Delta sdr_{ij}$ lies in the range $\left[ \frac{(k-1)\Delta sdr_{max}}{K}, \frac{k \cdot \Delta sdr_{max}}{K} \right]$. The output of the SI-SDR head of the network is a probability distribution over all $K$ classes. Similar to the Preference task, the network first produces frame-level distributions which are then temporally averaged to obtain the output $\mathbf{d}_{ij}$ for $\mathbf{x}_{ij}$.

The $K$ classes here are not entirely independent and share strong inter-class relationships. For example, the $k^{th}$ class is semantically much closer to $(k+1)^{th}$ class than, say $(k+4)^{th}$ class. This raises concerns over using the standard cross-entropy loss function with one-hot vectors as target labels. Moreover, training the network with the standard cross-entropy loss can lead to overconfident prediction and a lack of generalizations in unseen conditions [57]. This is especially a concern for us as we expect the network to learn the quality differences between non-matching inputs. Considering these factors, we propose to use *gaussian-smoothed-labels* [58] for computing the loss function. Label smoothing discounts certain probability mass from the true label and redistributes it to others. Formally, let $v$ be the class-index of the true class. The smoothed labels, $^s\mathbf{z}_{ij}$ are obtained as

$$^s\mathbf{z}_{ij}^k = \begin{cases} 0.6 & k=v \\ 0.2 & k=v\pm 1 \\ 0 & \text{otherwise} \end{cases} \qquad (4)$$

The same procedure is applied to the SNR head as well. The overall objective loss function for the quantification task is then defined as

$$L_Q(\mathbf{x}_{ij}, {}^s\mathbf{z}_{ij}) = L_{sdr}(\mathbf{x}_{ij}, {}^s\mathbf{z}_{ij}) + L_{snr}(\mathbf{x}_{ij}, {}^s\mathbf{u}_{ij}) = -\sum_{k=1}^{K} {}^sz_{ij}^k \log(d_{ij}^k) - \sum_{k=1}^{K} {}^su_{ij}^k \log(t_{ij}^k) \quad (5)$$

Similar to ${}^s\mathbf{z}_{ij}$ and $\mathbf{d}_{ij}$, ${}^s\mathbf{u}_{ij}$ and $\mathbf{t}_{ij}$ are the smoothed-label targets and probability outputs, respectively, for the SNR head. The total loss for training is $L = L_P + L_Q$.

### 3.4 Training Procedure

We now describe our training procedure. We assume the availability of a clean speech database $\mathcal{D}_c$. The training input $\mathbf{x}_{ij}$ is created by sampling two clean recordings $s_i$ and $s_j$ from $\mathcal{D}_c$. $s_i$ and $s_j$ are corrupted to produce $x_i$ and $x_j$. The degradation's we use can be largely grouped under two categories (a) additive noise degradations, and (b) speech distortions based on signal manipulations - *Clipping*, *Frequency Masking*, and *Mu-law compression*. For additive noise, we sample noise recordings, $n_i$ and $n_j$, from a noise database and add them to $s_i$ and $s_j$ at SNR levels uniformly sampled from the range -15dB to 60dB. This corresponds to a range of -15dB to +25dB in terms of SI-SDR. To ensure that SNR is consistent for $x_i$ and $x_j$, and for more stable training, the noise recordings $n_i$ and $n_j$ should generally belong to similar noise types. For speech distortions, we use the Audiomentations [2] toolkit. It allows one to select levels for various distortions and we select levels that always lie in the range of -15dB to +25dB. Once we have the degraded signals ($x_i$ and $x_j$) and their SNR and SI-SDR ($snr_{x_i}, snr_{x_j}, sdr_{x_i}, sdr_{x_j}$), we can train the network as described in previous sections. Note that, for degradations under the second category, the *quantification-task* loss $L_Q$, includes loss only from the SI-SDR head, $L_{sdr}$. SNR cannot be accurately computed for these distortions (residual $x - s$ may not be orthogonal to $s$) and hence we rely only on SI-SDR for training the network in these cases.

### 3.5 Usage

Once the network is trained, we can predict the quality of a test input $x_{test}$ with respect to any provided reference $x_{ref}$. As mentioned earlier, this reference need **not** be the matching clean reference. The quality score, NORESQA , of $x_{test}$ w.r.t $x_{ref}$ is obtained from the SI-SDR head of the network. It is obtained as

$$\text{NORESQA}_{x_{test}, x_{ref}} = \sum_{k=1}^{K} d_{x_{test}, x_{ref}}^k \mu^k, \quad \text{where} \quad \mu^k = \frac{1}{2}\left[\frac{(k-1)\Delta sdr_{max}}{K} + \frac{k \cdot \Delta sdr_{max}}{K}\right] \tag{6}$$

Here $d_{x_{test}, x_{ref}}^k$ and $\mu^k$ are the probability and the mean of the $\Delta sdr$ range of the $k^{th}$ class respectively. The NORESQA score gives us only the magnitude of difference between $x_{test}$ and $x_{ref}$. The output of the preference-task tells us the "sign", whether $x_{test}$ is better or worse than $x_{ref}$.

Our method provides a way to compare the quality of any two given speech recordings. However, in practice, one might often be interested in measuring "true" or "absolute" quality, which under our framework can be done by using clean or high-quality speech recordings as references. More specifically, the $x_{ref}$ samples should come from *any* clean speech database. Moreover, to reduce variance in the estimate, we can sample multiple references and obtain an average NORESQA score, $\text{NORESQA}_{x_{test}, x_{ref}}^{avg} = \frac{1}{n}\sum_{i=1}^{n} \text{NORESQA}_{x_{test}, x_{ref}^i}$, where $x_{ref} = \{x_{ref}^i\}\ \forall i \in [1, n]$ are $n$ NMRs sampled from the database.

## 4 Experimental Setup

**Datasets, Implementations and Baselines.** For training and validation, we choose the clean speech recordings from the DNS Challenge [59]. FSDK50 [60] serves as the noise dataset for additive noise degradations. Along with additive noise, clipping and frequency masking distortions are used during training. For robustness and better generalization to realistic conditions, we also add reverberation using room impulse responses from DNS Challenge dataset. For the test-set, we use TIMIT [61] as the source for clean speech, and ESC-50 [62] dataset for noise recordings. The test-set also includes Gaussian noise addition and Mu-law compression as unseen degradations.

---

[2] https://github.com/iver56/audiomentations

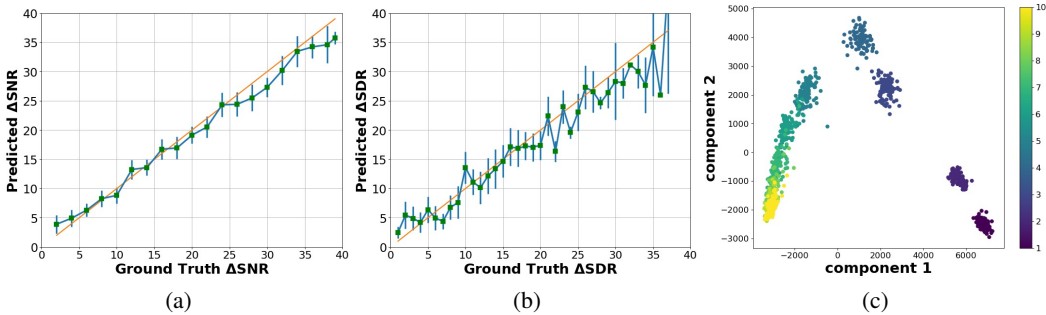

Figure 2: **(a) and (b)**: Average output of the model at different $\Delta snr$ and $\Delta sdr$. The vertical lines at each point in the plot shows 95% C.I. **(c)**: PCA visualization of embeddings capturing audio quality information.

Our method was implemented using Pytorch [63]. We use Adam [64] optimizer with a learning rate of $10^{-4}$ with a batch size of 32. We train the network for 1000 epochs using 4 Tesla V100 gpus. Please refer to supplementary material for further details on the datasets and implementations.

We use the well-known PESQ [2] and CDPAM [7] as full-reference baselines for SQA. CDPAM is a full-reference neural network based approach trained on human-pairwise judgements. Being full-reference, these methods require the paired (matching) clean reference for quality assessment. We use DNSMOS [1] as non-intrusive SQA method. It is trained on a large scale dataset of MOS ratings. However, no such large-scale dataset of MOS ratings is publicly available.

**Evaluation Methods.** We exhaustively evaluate our proposed framework in a variety of ways. The first series of evaluations, which we call *Objective Evaluations* for simplicity, focus on evaluating the models performance on each task (preference and quantification), and its in-variance to different NMR conditions, and so on. The second series of evaluations, referred to as *Subjective Evaluations*, applies our model to a series of speech problems and evaluates how well NORESQA is correlated to subjective ratings (e.g MOS). Lastly, we show the utility of our model in improving learnability in a downstream task, speech enhancement.

## 5 Results

### 5.1 Objective Evaluations

**Performance on Preference and Quantification Tasks.** The test inputs are created by randomly selecting two different clean recordings from the test set (TIMIT) and then degrading them as per the procedure in Sec. 3.4. We then measure the performance of the model on each task. On the Preference task, the model achieves an accuracy of *97.3%*.

The performance on the Quantitative tasks are shown in Fig 2(a) and (b) for $\Delta snr$ and $\Delta sdr$ respectively. On an average the model predicts close to the ground truth, $\Delta snr$ / $\Delta sdr$, for almost all cases. The standard deviations at most $\Delta snr$ / $\Delta sdr$ levels are small as well.

**In-variance to Language and Speaker's Gender.** We next evaluate the in-variance and robustness of NORESQA to certain characteristics of speech content such as language and speaker's gender. We find that our model is fairly robust to unseen languages. Moreover, for a given test recording, it does not matter whether the language or the speaker's gender in the reference is same or not. This supports our key hypothesis that non-matching references are sufficient for quality assessment. Please refer to supplementary material for details.

**Commutativity and Indiscernibility of Identicals.** We empirically study how well our framework supports these two desirable properties. **(1)** *Commutativity* - Quality assessment should stay same for $\mathcal{N}(x_1, x_2)$ and $\mathcal{N}(x_2, x_1)$. Changing the order of inputs should not change the NORESQA score. We find that for only a small fraction (less than *2.5%*) of the test pairs, changing the order also changes the NORESQA score by more than 2dB. Preference task outputs remains consistent as well (flips on changing the order) for more than *97%* of the test pairs. **(2)** *Indiscernibility of Identicals* - When both inputs to the models are same, $\mathcal{N}(x, x)$, the probability outputs of the preference task are close to *0.5* for most of the test cases. This is expected as the model is not able to confidently identify which input is cleaner. The NORESQA scores are also small in most cases, showing that the model supports this property to a considerable extent. Note that our learning mechanism does not explicitly enforce the framework to have these two properties. Hence, small errors are expected.

| Type | Name | VoCo [65] | | Dereverb [66] | | HiFi-GAN [67] | | FFTnet [68] | |
|------|------|-----|-----|-----|-----|-----|-----|-----|-----|
| | | PC | SC | PC | SC | PC | SC | PC | SC |
| Full-ref. | PESQ | 0.68 | 0.43 | **0.86** | 0.85 | 0.72 | 0.7 | 0.51 | 0.49 |
| | CDPAM | - | **0.73** | - | **0.93** | - | 0.68 | - | **0.68** |
| Non-Int. | DNSMOS | 0.6 | 0.48 | 0.7 | 0.73 | **0.93** | **0.88** | **0.59** | 0.53 |
| NORESQA | Paired | 0.64 | 0.6 | 0.46 | 0.65 | 0.59 | 0.81 | 0.46 | 0.47 |
| | Unpaired | 0.88±0.01 | 0.41±0.06 | 0.63±0.01 | 0.75±0.02 | 0.63±0.01 | 0.71±0.01 | 0.46±0.01 | 0.51±0.02 |
| | +Local-Fixed | **0.89±0.01** | 0.44±0.06 | 0.63±0.01 | 0.75±0.01 | 0.61±0.01 | 0.73±0.01 | 0.46±0.01 | 0.51±0.01 |
| | +Global-Fixed | 0.85±0.01 | 0.68±0.03 | 0.66±0.02 | 0.67±0.02 | 0.68±0.01 | 0.78±0.01 | 0.33±0.01 | 0.44±0.01 |

Table 1: **MOS Correlations (1)**: for NORESQA , DNSMOS, PESQ and CDPAM. Spearman (SC), Pearson (PC) correlations are shown. All unpaired NORESQA are obtained using $n = 100$ NMRs. ↑ is better.

| Type | Name | PEASS [69] | | VCC-2018 [70] | | Noizeus [71] | | TCD-VoIP [72] | |
|------|------|-----|-----|-----|-----|-----|-----|-----|-----|
| | | PC | SC | PC | SC | PC | SC | PC | SC |
| Full-ref. | PESQ | **0.86** | 0.71 | **0.51** | 0.56 | 0.43 | 0.42 | **0.89** | **0.90** |
| | CDPAM | - | **0.74** | - | **0.61** | - | **0.71** | - | 0.88 |
| Non-Int. | DNSMOS | 0.39 | 0.21 | 0.37 | 0.42 | 0.41 | 0.59 | 0.71 | 0.72 |
| NORESQA | Paired | 0.26 | 0.43 | 0.48 | 0.39 | 0.47 | 0.46 | 0.38 | 0.44 |
| | Unpaired | 0.38±0.01 | 0.40±0.01 | 0.61±0.01 | 0.41±0.02 | **0.50±0.02** | 0.39±0.05 | 0.43±0.01 | 0.46±0.02 |
| | +Local-Fixed | 0.40±0.04 | 0.52±0.06 | 0.65±0.04 | 0.39±0.02 | 0.45±0.01 | 0.44±0.02 | 0.43±0.02 | 0.41±0.04 |
| | +Global-Fixed | 0.41±0.05 | 0.57±0.05 | 0.47±0.01 | 0.41±0.01 | 0.48±0.02 | 0.51±0.01 | 0.56±0.01 | 0.52±0.03 |

Table 2: **MOS Correlations (2)**: for NORESQA , DNSMOS, PESQ and CDPAM. Spearman (SC), Pearson (PC) correlations are shown. All unpaired NORESQA are obtained using $n = 100$ NMRs. ↑ is better.

**Quality based Retrieval and Visualizations.** In this evaluation, we try to understand the representations learned by the network. More specifically, we consider the outputs of the temporal learning block as the Quality Embeddings (QE) and use it for visualizations and quality based retrieval. We first create a dataset of 1000 recordings at 10 different discrete quality levels (100 per quality level). In the quality based retrieval task, the goal is to retrieve recordings of same quality level for a given query speech recording. We argue that the QEs can be used to represent audio recordings for such a quality based retrieval. We use Precision@K to assess the top-$K$ retrievals. The mean precision@K over all test queries turns out to be *0.97* for $K = 10$ and *0.95* for $K = 25$. These high precision retrievals show that the embeddings indeed capture quality.

Further, Fig 2(c) shows a PCA visualization of the embeddings. We see that the separation between the clusters increases as the quality goes down. For higher qualities, the inter-cluster distances are smaller, although the embeddings at the same quality level are still tightly clustered together. Moreover, we also observe the cluster centers lie on approximately two piece-wise linear functions, one for low quality (1 to 4) and another for high quality (5 to 10). In other words, the model has implicitly learned high and low quality functions.

## 5.2 Subjective Evaluations

The subjective evaluations are aimed at assessing NORESQA as a proxy for subjective judgments by humans. More specifically, we try to understand how well NORESQA correlates with MOS across different speech tasks. We also examine NORESQA's suitability in subjective 2-alternative forced-choice (2AFC) tests [6].

We consider an exhaustive set of 10 different datasets for this evaluation. These datasets span over a variety of well-known speech problems; **(1)** Speech Synthesis (VoCo [65] and FFTnet [68]), **(2)** Speech Enhancement (Dereverberation [66], Noizeus [71], HiFi-GAN [67]), **(3)** Voice Conversion (VCC-2018 [70]), **(4)** Speech Source Separation (PEASS [69]), **(5)** Telephony Degradations [72], **(6)** Bandwidth Extension (BWE [73]), and **(7)** General Degradation's (Simulated [6]). Please refer to supplementary material for details about these datasets.

**MOS Correlations.** All of the above datasets come with MOS ratings for each audio recording. We evaluate our framework by computing Pearson Correlation Coefficient (PC) and Spearman's Rank Order Correlation (SC) of NORESQA scores with MOS ratings on each dataset. Since MOS scores are always obtained keeping a clean recording in mind, we obtain "absolute" NORESQA scores (Sec 3.5) by using clean references. As mentioned in Sec 3.5, we can get a more reliable score by averaging over multiple references. The results discussed in this section are for $n = 100$ and an ablation over $n$ are provided in further sections.

Finally, to decouple the effect of source/type of references on the results, we compute NORESQA for a given test recording in 4 different ways. **(i)** *Paired*: This is a special case where matched clean recordings are given as references to NORESQA. Obviously $n = 1$ in this case. **(ii)** *Unpaired*: In this case, for the given test recording, a clean NMR is randomly selected from the same dataset. The process is repeated $n = 100$ times and the scores are average to obtain the final NORESQA for the

| Name | Simulated [6] | FFTnet [68] | BWE [73] | HiFi-GAN [67] |
|---|---|---|---|---|
| PESQ | 86.0 | 67.0 | 38.0 | 88.5 |
| CDPAM | **87.7** | **88.5** | **75.9** | **96.5** |
| DNSMOS | 49.2 | 58.8 | 45.0 | 62.3 |
| NORESQA | 68.7 | 73.3 | 53.3 | 81.6 |

Table 3: Accuracy on 2AFC predictions

| Type | Data% | PESQ | STOI | SNRseg | CSIG | CBAK | COVL |
|---|---|---|---|---|---|---|---|
| Noisy | | 1.97 | 91.50 | 1.72 | 3.35 | 2.44 | 2.63 |
| Baseline | 33% | 2.22 | 91.7 | 8.18 | 3.26 | 2.98 | 2.72 |
| | 66% | 2.30 | 92.23 | 8.54 | 3.45 | 3.04 | 2.85 |
| | 100% | 2.39 | 91.89 | 8.71 | 3.55 | 3.10 | 2.95 |
| Pre-trained | 33% | 2.28 | 92.30 | 8.33 | 3.43 | 3.03 | 2.83 |
| | 66% | 2.35 | 92.90 | 8.77 | 3.53 | 3.1 | 2.92 |
| | 100% | **2.46** | **93.53** | **8.81** | **3.59** | **3.17** | **2.99** |

Table 4: Evaluation of NORESQA pre-training for speech enhancement.

test recording. **(iii)** *Unpaired Local-Fixed*: the same set of NMRs are used for all test recordings. The set ($n = 100$) is fixed locally for each dataset. **(iv)** *Unpaired Global-Fixed*: this is similar to the previous case, where the same set of NMRs are used for all test recordings. However, in this case, this set is fixed across all datasets. The NMRs ($n = 100$) are selected from the DAPS dataset [74] and used for evaluations on all 8 datasets. All unpaired experiments are repeated 10 times and average results with standard deviations are reported.

Tables 1 and 2 shows correlations with MOS on all 8 datasets. Correlations for full-reference SQA methods (PESQ and CDPAM), and non-intrusive DNSMOS are also shown. First, we note that NORESQA , is not only competitive to these methods but it even surpasses their performance in several cases. Our method is better than DNSMOS on 4 datasets (VoCo, PEASS, VCC and Noizeus) and is very competitive on others. Note that, DNSMOS is *explicitly* trained on a large scale MOS dataset. Our method is not trained on any perceptual labels or judgments and is still almost as good as DNSMOS. The results from the tables also show that our method is a good substitute for full-reference methods in several cases. However, unlike these full-reference methods, our framework stays useful in practical and real-world situations where the clean matching reference might not be available.

**2AFC Tests.** 2AFC test is a comparative approach to subjective evaluations. In this case, listeners are given a reference and two test recordings and asked to judge which one sounds more similar (in terms of quality) to the reference. We follow the evaluation protocol of Manocha et al. [6] and report how accurately different methods can predict same judgement as humans. Table 3 shows accuracy of different methods on 4 datasets. Full-reference methods give best performances in most cases. However, our framework, NORESQA is substantially better than DNSMOS in all cases.

**Ablations.** To further expound NORESQA , we conducted ablation studies for two factors. Please refer to the supplementary material for detailed results and analyses of these ablations.

*Multi-objective Learning of Quantification Task*: In this ablation, we try to assess the significance of SNR and SI-SDR heads for the Quantification task in NORESQA. Our ablation shows that using both heads is superior to using just SNR or SI-SDR head; often the improvement is more than 30 to 40% over using just one of the heads. SI-SDR is also performs better in all cases when it comes to having just SNR or SI-SDR for the quantification task. This is expected, as with SI-SDR, we can use a wide variety of degradations, whereas SNR based training can only use additive noise degradations.

*Number of NMRs ($n$)*: In this ablation, we examine how the MOS correlations are affected by the number of NMRs used for each test recording. Increasing the number of NMRs for NORESQA improves correlations with MOS. Depending on the dataset, this improvement can be as much as 10 to 15% when $n$ is increased from 1 to 100.

## 5.3 Speech enhancement

In this section, we show the utility of NORESQA for training a Speech Enhancement (SE) system. Our SE network is based on a popular U-Net type convolutional recurrent neural network [75]. The baseline SE model is trained in a supervised manner using matched clean and noisy speech pairs. Note that the idea behind using NORESQA for SE is quite different from various works that use a speech enhancement model to guide training [76]. These methods rely on metrics to compute a loss function (generally in addition to L1/L2 losses w.r.t the target clean speech). They train local differentiable proxies of quality (e.g., PESQ) or intelligibility (e.g., STOI) at every iteration of training. Clearly, this requires matching noisy-clean pairs for training. MetricGAN does this in a GAN framework for improving the discriminator. Our approach for using NORESQA for SE is very different. Since NORESQA can compare two non-matched recordings, we use it to pre-train the Speech Enhancement model in a way that does not require the exact matched pairs of recordings. This can be much more useful for out-of-domain data adaptation, unseen noise conditions, and other sparse labeled-data situations. Our approach can leverage potentially large amounts of unpaired data in these cases. Moreover, as already mentioned in Sec 1, using certain metrics such as PESQ as

the target quality has its own issues, including sensitivity to perceptually invariant transformations especially in cases where the quality between two recordings to be compared are perceptually close.

We show that pre-training the network using NORESQA as loss function can lead to improvements in performance. More specifically, given a noisy speech recording and a non-matching clean speech recording, during pre-training stage, the SE model tries to minimize the NORESQA score between the output of the SE model and clean recording. Since we do not require *any* matched clean and noisy pairs for this step, we can leverage unlimited amount of training data - most significantly noisy recordings from real world for which the corresponding clean recordings are not available. After the pre-training stage, the model can be fine-tuned for enhancement following the usual paired training strategy. We use the VCTK dataset [77] for these experiments and consider 3 baseline SE models. These are models trained on $1/3^{rd}$, $2/3^{rd}$ and full training set. Following prior works on this dataset, we report enhancement performance on 5 different objective metrics.

Table 4 shows the benefits of NORESQA based pre-training. We see that pre-training consistently improves all 5 metrics for all 3 baseline models. We further analyse the improvements at different SNR levels, and observe that NORESQA based pre-training is most effective in high SNR conditions, where the degradation caused by strong background noise is not noticeable and harder to learn. Please refer to the supplementary material for further details. Our motivation behind these experiments is to demonstrate proof of concept of NORESQA in an application (e.g., SE). Hence, our focus is not on obtaining SOTA performance on the VCTK dataset. The improvements coming from pre-training with NORESQA show the potential of leveraging the differentiable NORESQA in different downstream tasks.

## 6 Conclusion

In this paper, we presented a new framework for speech quality assessment. Motivated by human's ability to assess quality independent of the speech content, we propose a framework for SQA using non-matching references. This can potentially open up a new research direction in the development of SQA methods. In subjective evaluations, our method works as competently as established full-reference and subjective non-intrusive SQA methods. At the same time, it addresses key limitations of those methods. Going forward, our focus will be on developing novel methods under the NORESQA framework which can correlate better with subjective human ratings.

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
