# OpenReview forum: "NORESQA: A Framework for Speech Quality Assessment using Non-Matching References"
_NeurIPS.cc/2021/Conference — NeurIPS 2021 Poster_

### Official Review · Reviewer_fWQh · 2021-07-16

**Rating:** 6
**Confidence:** 4

**Summary:**

This paper describes a new approach to speech quality estimation, that of using a non-matching reference. Instead of either using the exact clean speech that was used to produce a noisy or corrupted utterance, as "intrusive" metrics do, or not using any reference, as "non-intrusive" metrics do, it performs a comparison with a potentially unrelated utterance presumably in terms of the overall quality and nothing else. Thus it has the advantage of in some sense being a relative comparison for reduced variance, while also not requiring synthetic mixtures, so can be widely applied in the real world. By comparing one test utterance to many different unrelated references and averaging the scores together, the quality can be estimated more precisely. On evaluations on a number of datasets with subjective ratings, the proposed approach is more correlated with human judgments than the non-intrusive "DNSMOS" on about as many as it is less correlated, both of which are less correlated than PESQ and CDPAM. In using the model to make forced choices, it agrees with human choices more often than DNSMOS, but less the PESQ and CDPAM. Since the metric is differentiable, it can be used to train a speech enhancement system. In experiments using it to pre-train a model, it improves performance by 0.07 MOS for PESQ and 0.016 STOI points (out of 1).

**Limitations And Societal Impact:**

Limitations and broader societal impacts are not discussed by the paper, except that the authors do state in the checklist that they foresee no negative societal impacts, with which I agree.

**Main Review:**

This paper presents an interesting and novel angle on speech quality estimation. It seems a little bit crazy at first, but ends up making sense. Even though a given reference can't be used to compare directly with the utterance under test, features of it, especially in terms of its quality, can be. The idea of averaging over a small corpus of such references makes the idea make more sense and provides interesting opportunities to decide what that reference corpus should be.

One issue with the paper is that the proposed approach is motivated as being preferrable to the non-intrusive DNSMOS because it is more adaptable to new datasets and can avoid domain shift from the (large) labeled dataset that DNSMOS was trained on. None of the datasets identified in the paper, however, appear to meet this criterion, so I wonder how much of a problem it really is for DNSMOS. The experiments could perhaps be adjusted to highlight the strengths of the proposed approach.

There are a couple of ways in which the model has been formulated that may be making its job harder. In particular, it takes as input both magnitude and phase, but it is hard for a deep network to use phase directly, even to make comparisons, because of the phase wrapping issue. I would recommend using cos(phase) and sin(phase) instead, which facilitate comparisons both across time and between utterances. In addition, the model is trained to predict the sign and the absolute difference in quality between the two signals. It seems like it would be easier for the model to learn to predict the signed difference.

The paper uses nearly the same equation to define SNR and SI-SDR ((1) vs (2)). I believe that the SI-SDR implementation is not only an instantaneous subtraction, but uses a least squares estimate of a transfer function to cancel out the signal from the noise. It is not clear whether the implementation of SNR does this or not. This should be clarified in the equations and if they are really so similar, then why not call them SDR and SI-SDR?

Overall, this is an interesting and novel approach, although the stated advantages of the approach are not entirely borne out by the experiments.

**Time Spent Reviewing:**

2

---

> ### Author Response · Authors · 2021-08-10
> **Response to Reviewer fWQh**
>
> We thank the reviewer for their constructive feedback and suggestions. Next, we would like to address the specific comments of the reviewer.
>
> 1. ***One issue with the paper is that the proposed approach is motivated as being preferable to the non-intrusive DNSMOS because it is more adaptable to new datasets and can avoid domain shift from the (large) labeled dataset that DNSMOS was trained on. None of the datasets identified in the paper, however, appear to meet this criterion, so I wonder how much of a problem it really is for DNSMOS. The experiments could perhaps be adjusted to highlight the strengths of the proposed approach.***
>
> Our primary motivation for NORESQA being preferable over DNSMOS is not adaptability to domain shift, although NORESQA does generalize well to out-of-domain datasets as well. The major advantage is that, unlike DNSMOS, NORESQA does not rely on manually obtained MOS ratings. DNSMOS or any other approach relying on MOS are trained on a large-scale dataset with speech recordings labeled with their MOS by humans. As mentioned in the paper (page 2, paragraph 1), collecting such a dataset can be prohibitively resourced hungry and a tedious process. Consistency in audio testing infrastructure (hardware, speakers, headphones, etc), listening environments, uniformity among thousands of raters, their hearing profile, etc. is necessary to get reliable scores. The dataset of DNSMOS has high label noise which makes it difficult to learn from such a dataset as well.
>
> NORESQA on the other hand is an **unsupervised approach in that it does not rely on manually obtained MOS ratings**. For the MOS correlation task, even though NORESQA is not trained directly on MOS, it performs competitively with DNSMOS on several datasets and even outperforms on several. Several of these datasets are out-of-domain for NORESQA and as well as DNSMOS,  the distortions are different from those used during training.
>
> Lastly, in the second subjective evaluation through the 2AFC tests,  NORESQA consistently outperforms DNSMOS showing that DNSMOS does not generalize well to subjective evaluation methods beyond MOS whereas NORESQA does. This shows that NORESQA is actually learning generalizable quality attributes which can be used in a variety of subjective tests, beyond MOS whereas DNSMOS and other such systems might be too focused on a single metric.
>
> 2. ***There are a couple of ways in which the model has been formulated that may be making its job harder. In particular, it takes as input both magnitude and phase, but it is hard for a deep network to use phase directly, even to make comparisons, because of the phase wrapping issue. I would recommend using cos(phase) and sin(phase) instead, which facilitate comparisons both across time and between utterances.***
>
> We appreciate the reviewer’s comment and suggestion on learning from phase. We agree that sometimes it can be difficult for deep neural networks to learn efficiently from phase. While it might be possible that the neural network can learn the sine and cosine functions, a better approach to incorporate phase might indeed be helpful. We will explore methods on how to better incorporate the phase for quality score and report it in future works.
>
> 3. ***In addition, the model is trained to predict the sign and the absolute difference in quality between the two signals. It seems like it would be easier for the model to learn to predict the signed difference.***
>
> We would like to add that we had tried the suggested approach and found that it is suboptimal in comparison to our approach. Predicting the two outputs offers a few advantages. The primary reason why separating the learning process into separate tasks, a preference and quantification task, is important is because the inputs are non-matching pairs. If we use signed intervals, it might be non-trivial for the model to learn to discard speech content differences and learn quality features. Disentangling the learning through two separate tasks makes it easier for the mode to focus on quality attributes. The errors on the two tasks are not completely intertwined. Another advantage, in this case, is that the architecture of the two output paths can also be appropriately adjusted if necessary to improve performance.
>
> Moreover, it also makes the model easier to use - in the sense that if only preference results are needed or if only absolute quality difference is required, the appropriate head can be used while discarding the unused one. Lastly, the framework can be easily extended through the separate preference and quantification tasks. For example, we can have more than 2 inputs and the preference task can be designed to predict a ranked order in terms of quality.
>
> 4. ***The paper uses nearly the same equation to define SNR and SI-SDR ((1) vs (2)). I believe that the SI-SDR implementation is not only an instantaneous subtraction, but uses a least squares estimate of a transfer function to cancel out the signal from the noise. It is not clear whether the implementation of SNR does this or not. This should be clarified in the equations and if they are really so similar, then why not call them SDR and SI-SDR?***
>
> Eq 1 and Eq 2 in the paper are different. The reviewer is correct in suggesting that SI-SDR involves least square estimation and our equation for SI-SDR does indicate that.
> We use the term SNR for only additive noise degradations and in this case, SNR and SDR would be the same. We use the term ‘SNR’ just to focus on additive noise since it is more common to refer to them as SNR-based distortions, rather than SDR-based distortions.

---

> > ### Author Response · Authors · 2021-08-20
> > **Concerns addressed ?**
> >
> > Hello reviewer fWQh,
> > We would be grateful if you can confirm that our response has addressed your concerns ? Please let us know if any issues remain.

---

> > ### Comment · Reviewer_fWQh · 2021-08-31
> > **Thank you for the clarifications**
> >
> > I thank the authors for their responses. Below I will reply to each one using the same numbers.
> >
> > 1. While I agree that not requiring a large human-annotated corpus of MOS ratings is a potential advantage of the proposed approach over DNSMOS, the fact is that such a corpus has already been collected, so whether it is required or not is a bit of a moot point. This would potentially not apply to other conditions that are not included in the DNSMOS dataset, however, which is why it would be good to show that the proposed approach out performs DNSMOS on such conditions.
> >
> > 2. Sounds good.
> >
> > 3. Thank you for clarifying this. Perhaps this is worth mentioning in the paper.
> >
> > 4. My question was whether SI-SDR in addition to using a least squares estimate of the amplitude, also performs a least squares estimate of an entire convolutional transfer function, which is what the BSS_EVAL toolbox uses. This allows SDR to be robust to simple delays, while SNR requires strict time alignment to the sample. I don't see anything in the notation for equations 1 and 2 that indicates that one of them is convolutional. Please clarify this in the paper.

---

> > > ### Author Response · Authors · 2021-08-31
> > > **Further clarifications to Reviewer fWQh**
> > >
> > > We thank the reviewer for the response. Below, we would like to reply to each point:
> > >
> > > 1. ***Regarding DNSMOS Dataset***: Please note that the dataset of MOS ratings used in the DNSMOS paper is not open source and we don’t know if it will be released at all. To the best of our knowledge, there is no large-scale publicly available MOS dataset for training DNSMOS and such models, which clearly hinders research in this area. Secondly, unsupervised approaches such as ours – which do not rely on large labeled datasets - are significant even if such a dataset existed as supervised learning have limitations. As pointed out by other reviewers, our unsupervised approach is likely applicable to other situations as well.\
> > > We have indeed considered degradations that are outside of the domain of DNSMOS, e.g. PEASS and VCC-2018. DNSMOS does not achieve good correlations on these datasets (Table 2) whereas our method obtains much better correlations. Moreover, on subjective evaluations beyond MOS (2AFC tests), our method outperforms DNSMOS by a considerable margin (Table 3). These results show that NORESQA not only generalized better than DNSMOS in out-of-domain datasets but also on perceptual tests beyond MOS.
> > >
> > > 4. ***SI-SDR formulation***: Thank you for pointing it out. Our definition of SI-SDR follows Le Roux et al.[1] which doesn’t use the convolutional transfer function normalization. We would point it out in the paper.
> > >
> > > [1] SDR – HALF-BAKED OR WELL DONE? Jonathan Le Roux, Scott Wisdom, Hakan Erdogan, John R. Hershey https://arxiv.org/pdf/1811.02508.pdf

---

### Official Review · Reviewer_i3ZZ · 2021-07-17

**Rating:** 7
**Confidence:** 5

**Summary:**

This submission presents a novel framework for speech quality assessment using non-matching references (NORESQA). The main idea is to use relative scale-invariant signal to distortion ratio (SI-SDR) and signal-to-noise ratio (SNR) scores to form training targets. NORESQA is then trained in a multi-task learning criterion and aims to produce speech quality assessment scores. A major advantage of NORESQA is that it does not require matched clean reference when calculating assessment scores. The idea of using the relative SNR and SI-SDR scores to form the training targets is novel and convincing. The promising results confirm the effectiveness of the proposed approach.

**Ethical Concerns:**

We do not find any ethical concerns in this submission.

**Limitations And Societal Impact:**

(1) The performance of the speech enhancement on VCTK dataset is not competitive against the state-of-the-art systems (https://paperswithcode.com/sota/speech-enhancement-on-demand), making it difficult to verify the effectiveness of applying NORESQA to pretrain a speech enhancement system. A stronger speech enhancement baseline should be used.
(2) The proposed model can be used in various speech-related applications.

**Main Review:**

This submission presents a novel framework for speech quality assessment using non-matching references (NORESQA). The main idea is to use relative scale-invariant signal to distortion ratio (SI-SDR) and signal-to-noise ratio (SNR) scores to form training targets. NORESQA is trained in a multi-task learning criterion and aims to produce speech quality assessment scores. A major advantage of NORESQA is that it does not require matched clean reference when calculating assessment scores. The idea of using the relative SNR and SI-SDR scores to form the training targets is novel and convincing. The promising results confirm the effectiveness of the proposed approach. We are providing our comments as below:
(1) Since the target of this work is to predict a continuous score, it is intuitive to use a regression model to build the assessment system. The authors in [1] have compared the mean opinion scores (MOS) predictions based on regression and classification models. The current submission only adopts a classification model to build the system. Please justify why a classification model is selected rather than a regression model.
(2) When performing quality score quantification, an equal division of 0 to max(SI-SDR difference) is used. Have you considered unequal divisions or learnable divisions?
(3) In [2, 3], a speech assessment model has been used to guide speech enhancement training. Please provide discussions on the difference between your works (applying NORESQA for speech enhancement) and the existing works [2, 3].
(4) The authors state that: “The degradation’s we use can be largely grouped under two categories (a) additive noise; (b) signal manipulations. In the experiments, the authors selected 8 datasets for evaluations. In some datasets, such as VoCo, FFTnet, and VCC-2018, no speech distortions are caused by additive noise and signal manipulations. It is interesting to further explore why the proposed NORESQA can perform well on these datasets. The authors may also want to provide analyses of SNR and SI-SDR scores obtained by the utterances in these datasets.
(5) The performance of the speech enhancement on VCTK dataset is not competitive against the state-of-the-art systems (https://paperswithcode.com/sota/speech-enhancement-on-demand), making it difficult to verify the effectiveness of applying NORESQA to pretrain a speech enhancement system. A stronger speech enhancement baseline should be used.



[1] Lo, C. C., Fu, S. W., Huang, W. C., Wang, X., Yamagishi, J., Tsao, Y., & Wang, H. M. (2019). MOSNet: Deep learning-based objective assessment for voice conversion. arXiv preprint arXiv:1904.08352.
[2] Fu, S. W., Liao, C. F., Tsao, Y., & Lin, S. D. (2019, May). Metricgan: Generative adversarial networks based black-box metric scores optimization for speech enhancement. In International Conference on Machine Learning (pp. 2031-2041). PMLR.
[3] Fu, S. W., Yu, C., Hsieh, T. A., Plantinga, P., Ravanelli, M., Lu, X., & Tsao, Y. (2021). MetricGAN+: An Improved Version of MetricGAN for Speech Enhancement. arXiv preprint arXiv:2104.03538.



**Time Spent Reviewing:**

10 hours

---

> ### Author Response · Authors · 2021-08-10
> **Response to Reviewer i3ZZ**
>
> We would like to thank the reviewer for the positive review and the feedback/comments. Next, we would like to address the specific comments of the reviewer.
>
> 1. ***Since the target of this work is to predict a continuous score, it is intuitive to use a regression model to build the assessment system. The authors in [1] have compared the mean opinion scores (MOS) predictions based on regression and classification models. The current submission only adopts a classification model to build the system. Please justify why a classification model is selected rather than a regression model.***
>
> A classification model in this setting offers several advantages over a regression model. A regression loss such as minimizing MSE, in this case, is less robust due to prediction outliers and may make the model prone to overfitting [2,3]. Given that our model is designed for non-matching inputs,  this becomes an even bigger concern and considerably affects the generalization capabilities of the model.  A classification framework helps reduce the risk of model overfitting, and large estimation outliers. Moreover, instead of forcing the output to predict one number through a regression model, the classification approach also offers flexibility in designing losses and learning formulations. These flexibilities allow us to learn better in our setting of non-matching inputs. We use label distribution learning [4] by making use of the ordering of the labels, as well as enforce inter-class label relationships, which is not in the conventional formulation of a regression model, or even the cross-entropy loss.
>
> 2. ***When performing quality score quantification, an equal division of 0 to max(SI-SDR difference) is used. Have you considered unequal divisions or learnable divisions?***
>
> We did try some uneven (e.g., log-spaced intervals - small gaps for low differences, and large gaps for large differences) intervals. However, we found that these uneven intervals do not, in general, yield consistent performance across all datasets. Moreover, by changing K, one can easily change the granularity of classes while keeping the system simple. It also leads to consistent performances across different datasets. Learnable divisions are indeed an interesting idea and we have not yet explored them. We will investigate it in future works.
>
> 3. ***In [5,6], a speech assessment model has been used to guide speech enhancement training. Please provide discussions on the difference between your works (applying NORESQA for speech enhancement) and the existing works [5,6]***
>
> The idea behind using NORESQA for Speech Enhancement (SE) is quite different from the referenced works. These methods rely on metrics to compute a loss function (generally in addition to L1/L2 losses w.r.t the target clean speech). They train local differentiable proxies of quality (e.g., PESQ) or intelligibility (e.g., STOI) at every iteration of training. Clearly, this requires matching noisy-clean pairs for training. MetricGAN does this in a GAN framework for improving the discriminator.
> Our approach for using NORESQA for SE is very different. Since NORESQA can compare two non-matched recordings, we use it to pre-train the Speech Enhancement model in a way that does not require the exact matched pairs of recordings. This can be much more useful for out-of-domain data adaptation, unseen noise conditions, and other sparse labeled-data situations. Our approach can leverage potentially large amounts of unpaired data in these cases and can lead to improvements in SE systems, as shown in our experiments.
>
> Moreover, using certain metrics such as PESQ as the target quality has its own issues.  It has been shown to be sensitive to perceptually invariant transformations [7,8], especially in cases where the audio quality between two recordings to be compared are perceptually close [9]. However, SNR and SI-SDR on which NORESQA is based can be considered as two of the most fundamental measures to quantify the quality of a signal. In fact, they have also been shown to have better properties than differences computed through conventional Euclidean distance [10].
>
> 4. ***The authors state that: “The degradation’s we use can be largely grouped under two categories (a) additive noise; (b) signal manipulations. In the experiments, the authors selected 8 datasets for evaluations. In some datasets, such as VoCo, FFTnet, and VCC-2018, no speech distortions are caused by additive noise and signal manipulations. It is interesting to further explore why the proposed NORESQA can perform well on these datasets. The authors may also want to provide analyses of SNR and SI-SDR scores obtained by the utterances in these datasets.***
>
> The motivation behind using a variety of datasets for different problems is to illustrate the generalization capabilities of our model on distortions that might be different from those used for training. FFTnet [11] has certain noisy artifacts (e.g., clicking), as well as speech distortions commonly found in speech synthesis. VCC2018 [12] also has a lot of different artifacts where one compares the naturalness of VCC samples compared to natural speech. VoCo [13], there are a few frame-level artifacts of where the synthesized frame is inserted, which is captured by our model. NORESQA performs well on these datasets showing that our training process ( with the two broad sets of degradations) used can generalize well on these datasets. We thank the reviewer for recommending to analyze the SNR and SI-SDR scores on these datasets. We will explore it and report any interesting findings in the updated/camera-ready version of the paper.
>
> 5. ***The performance of the speech enhancement on VCTK dataset is not competitive against the state-of-the-art systems (https://paperswithcode.com/sota/speech-enhancement-on-demand), making it difficult to verify the effectiveness of applying NORESQA to pretrain a speech enhancement system. A stronger speech enhancement baseline should be used.***
>
> As mentioned by multiple reviewers, the proposed model can be used in various speech-related applications. Our motivation behind these experiments is to demonstrate the use of NORESQA in one such application. Hence, our focus is not on obtaining SOTA performance on the VCTK dataset. We show that NORESQA can be used as a pre-training strategy for speech enhancement.
>
> As far as using a stronger speech enhancement baseline is concerned, the performance gap is likely coming because of our small model size. The best-performing model in the suggested link has around 10M parameters, whereas our model is almost an order of magnitude smaller with 1.1M parameters.
>
> Moreover, while VCTK is often used for evaluation, the training itself is done on much larger datasets whereas we are training only on VCTK which is a relatively small dataset. The improvements coming from pre-training with NORESQA show the potential of leveraging the differentiable NORESQA in different downstream tasks. For speech enhancement specifically, it shows that a model can be taught speech quality characteristics through NORESQA. It is not our intention to claim that NORESQA alone would outperform all other perceptually motivated losses in literature.
>
> [1] C. C. Lo et al. (2019). MOSNet: Deep learning-based objective assessment for voice conversion.
>
> [2] X. Dong et. al. A classification-aided framework for non-intrusive speech quality assessment in IEEE WASPAA, 2019
>
> [3] Y. Meng, et. al., MetricNet: Towards Improved Modeling For Non-Intrusive Speech Quality Assessment in Interspeech 2021
>
> [4] X. Geng, Label Distribution Learning: IEEE Transactions on Knowledge and Data Engineering
>
> [5] S. W. Fu et al. (2019, May). Metricgan: Generative adversarial networks based black-box metric scores optimization for speech enhancement. In International Conference on Machine Learning (pp. 2031-2041). PMLR.
>
> [6] S. W. Fu et al. (2021). MetricGAN+: An Improved Version of MetricGAN for Speech Enhancement. arXiv preprint arXiv:2104.03538
>
> [7] A. Hines, et.al., Robustness of speech quality metrics to background noise and network degradations: Comparing visqol, pesq and polqa in IEEE ICASSP 2013
>
> [8] T. Manjunath, Limitations of perceptual evaluation of speech quality on voip systems in 2009 IEEE ISBMSB
>
> [9] P. Manocha, et. al. A Differentiable Perceptual Audio Metric Learned from Just Noticeable Differences in Interspeech 2020
>
> [10] T. Yuan, Signal-to-Noise Ratio: A Robust Distance Metric for Deep Metric Learning in CVPR 2020
>
> [11] Z. Jin et. al. FFTNET: A REAL-TIME SPEAKER-DEPENDENT NEURAL VOCODER in ICASSP 2018
>
> [12] J.L. Trueba et. al.  The Voice Conversion Challenge 2018: Promoting Development of Parallel and Nonparallel Methods
>
> [13] Z. Jin et al. VoCo: Text-based Insertion and Replacement in Audio Narration in ACM Transactions on Graphics 2017

---

### Official Review · Reviewer_S2km · 2021-07-20

**Rating:** 8
**Confidence:** 3

**Summary:**

This paper presents a novel method to estimate speech quality. The key idea is to compare an incoming sample to a small database of existing samples and rate it as better or worse than each of the samples and by how much.
This is relative, or comparative approach contrasts with other methods which produce a global non-relative score.
This main contributions are a special loss function and a self-supervised training procedure that relies on signal corruption (and hence assumes that inputs are clean).
The estimator is implemented via convolutional nets that take 3 seconds of audio features.
The authors show that the score correlates well with both objective and perceptual scores.
The authors also show the usefulness of the loss function for pretraining of a speech-enhancement system.

**Ethical Concerns:**

I do not have any ethical concerns about this work.

**Limitations And Societal Impact:**

I do not see any obvious potential negative societal impact in this work.

**Main Review:**

This idea of a relative quality signal is quite original and a natural one .
I believe this approach will be applicable in a range of situations.
The presentation is clear and the experimental section seems robust: showing that the score correlates well with difference measures.

**Time Spent Reviewing:**

2

---

> ### Author Response · Authors · 2021-08-10
> **Response to Reviewer S2km**
>
> We would like to thank the reviewer for the positive review and the encouraging comments.

---

### Decision · Program_Chairs · 2021-09-27

**Decision:**

Accept (Poster)

**Comment:**

Overview: The paper presents a novel and simple method to measure quality of speech. They compare the input against a small set of examples. The comparison is performed used measures such as SNR and SI-SDR which do not require matching references. This is a big advantage for this sort of measurement. The experimental results bolster the claims in the paper.

Reviews: The reviewers were consistent in appreciating the novelty and simplicity of the proposed approach. The few minor nits and reservations that reviewers shared were adequately addressed by the authors in their response. I would urge the authors to incorporate them into the revised version of the paper.